# Body Weight, BMI, Percent Fat and Associations with Mortality and Incident Mobility Limitation in Older Men

**DOI:** 10.3390/geriatrics6020053

**Published:** 2021-05-18

**Authors:** Peggy M. Cawthon, Stephanie L. Harrison, Tara Rogers-Soeder, Katey Webber, Satya Jonnalagadda, Suzette L. Pereira, Nancy Lane, Jane A. Cauley, James M. Shikany, Samaneh Farsijani, Lisa Langsetmo

**Affiliations:** 1Research Institute, California Pacific Medical Center, San Francisco, CA 94143, USA; stephanie.harrison@ucsf.edu (S.L.H.); Katey.Webber@ucsf.edu (K.W.); 2Department of Epidemiology and Biostatistics, University of California San Francisco, San Francisco, CA 94158, USA; 3Sacramento VA Medical Center, Mather, CA 95655, USA; tsrogers@ucdavis.edu; 4Abbott Nutrition Inc., Columbus, OH 43219, USA; satya.jonnalagadda@abbott.com (S.J.); Suzette.Pereira@abbott.com (S.L.P.); 5Department of Medicine, University of California at Davis Medical Center, Sacramento, CA 95871, USA; nelane@ucdavis.edu; 6Graduate School of Public Health, University of Pittsburgh, Pittsburgh, PA 15261, USA; JCauley@edc.pitt.edu (J.A.C.); samaneh.farsijani@pitt.edu (S.F.); 7Division of Preventative Medicine, School of Medicine, University of Alabama at Birmingham, Birmingham, AL 35205, USA; jshikany@uabmc.edu; 8Division of Epidemiology and Community Health, University of Minnesota, Minneapolis, MN 55454, USA; langs005@umn.edu

**Keywords:** obesity, mortality, mobility limitation

## Abstract

How different measures of adiposity are similarly or differentially related to mobility limitation and mortality is not clear. In total, 5849 community-dwelling men aged ≥65 years (mean age: 72 years) were followed mortality over 10 years and self-reported mobility limitations (any difficulty walking 2–3 blocks or with climbing 10 steps) at six contacts over 14 years. Baseline measures of adiposity included weight, BMI and percent fat by DXA. Appendicular lean mass (ALM, by DXA) was analyzed as ALM/ht2. Proportional hazards models estimated the risk of mortality, and repeated measures generalized estimating equations estimated the likelihood of mobility limitation. Over 10 years, 27.9% of men died; over 14 years, 48.0% of men reported at least one mobility limitation. We observed U-shaped relationships between weight, BMI, percent fat and ALM/ht2 with mortality. There was a clear log-linear relationship between weight, BMI and percent fat with incident mobility limitation, with higher values associated with a greater likelihood of mobility limitation. In contrast, there was a U-shaped relationship between ALM/ht2 and incident mobility limitation. These observational data suggest that no single measure of adiposity or body composition reflects both the lowest risk of mortality and the lowest likelihood for developing mobility limitation in older men.

## 1. Introduction

Weight, body mass index (BMI) and percent body fat are inter-related measures that reflect overall body size and adiposity and are used to operationalize obesity. Obesity is very common in older adults in the United States [1] and is thought to be a powerful contributing factor in many age-related chronic health conditions, loss of functional capacity and mortality [2]. Some studies [3,4,5,6,7], but not all [8,9], have demonstrated a protective effect of overweight and obesity on mortality and other health outcomes in older adults; and some have demonstrated a U-shaped relationship with the highest risks in the lowest and greatest BMI. This relationship, wherein obesity (defined by high BMI) appears protective against mortality, has been termed the “obesity paradox” [10]. However, this apparently paradoxical association may be explained by the confounding influences of smoking and involuntary disease-associated weight loss; when these factors are taken into account, obesity or overweight have no longer been associated with higher rates of mortality [9]. Observational studies of older adults have also demonstrated an association between weight loss and increased risk of mortality [11,12,13]. However, the association between weight loss and increased risk of mortality was not supported by a meta-analysis of randomized trials of weight-loss interventions in older adults. [14]. Obesity is also widely considered to be a risk factor for mobility limitations [15].

In addition, there is also concern about the use of BMI as an approximation of the total adiposity burden in older adults. Since height is lost with age [16], the accuracy of BMI as a surrogate measure of adiposity may be worse in older than younger adults [17]. Whether a more direct measure of adiposity, such as percent body fat, is similarly related to both mortality and mobility limitations is relatively understudied: while previous studies have shown associations between high percent body fat and mobility limitations [18,19,20], mixed results of the association between body fat percentage and mortality have been reported [21,22,23,24]. Further, the relation between another major component of body composition, appendicular lean mass and mortality and mobility limitation has been reported, with lower lean mass often associated with increased mortality risk but having no influence on the risk of mobility limitation [25,26].

Therefore, we aimed to describe the relationships between weight (and BMI, percent body fat or lean mass) with mortality and mobility limitation over a long-term follow-up in a cohort of older, community-dwelling men. We hypothesized that once covariates were considered, weight, BMI and percent body fat would demonstrate a U-shaped relationship with both mortality and incident mobility limitation and that appendicular lean mass would be modestly related to mortality but unrelated to incident mobility limitation. We tested these hypotheses using data from the Osteoporotic Fractures in Men (MrOS) study. 

## 2. Materials and Methods

In 2000–2002, MrOS initially enrolled 5994 community-dwelling men aged 65 years and older, who were able to walk without the assistance of another person and free from bilateral hip replacements, at 6 academic medical centers in the United States as described previously [27,28]. Surviving participants were contacted to complete questionnaires and exams every 2–3 years over the subsequent 14.4 years. The study was approved by IRBs at all study sites, and all participants provided written informed consent.

### 2.1. BMI, Weight, Percent Fat and Total Energy Intake 

Weight was measured by balance beam scales at the baseline exam, height by wall-mounted stadiometers, and BMI was calculated as weight (kg)/height^2^ (m^2^). Body fat percent and appendicular lean mass were measured using whole-body dual-energy x-ray absorptiometry (DXA) scans on Hologic 4500 scanners (Hologic, Inc., Marlborough, MA, USA). A central quality control lab, certification of DXA operators, and standardized procedures for scanning were used to ensure the reproducibility of DXA measurements. At baseline, a Hologic whole-body phantom was circulated and measured at the 6 clinical sites. The variability across clinics was within acceptable limits, and cross-calibration correction factors were not required. Dietary data were derived from the brief Block 98 semi-quantitative food frequency questionnaire (FFQ) administered at baseline. The Block 98 FFQ has been used extensively and has been validated with diet records [29,30]. Participants with implausible values for total energy intake (<500 kilocalories per day) were excluded.

### 2.2. Mobility Limitation

At baseline and again at Year 5, 7, 9, 14 and 17 follow-up contacts, participants were queried about their ability and level of difficulty walking 2–3 blocks or climbing 10 stairs. Men who reported any difficulty with these measures were considered to have a mobility limitation at that time-point.

### 2.3. Mortality

Men were contacted by mail or follow-up phone call every 4 months after the baseline exam to query about recent falls and fractures. Clinic staff were usually notified of a participant’s death when following up on missed contacts. Deaths were centrally adjudicated by a physician review of the death certificates and hospital discharge summaries (when available); we included deaths that occurred within 10 years of the baseline exam.

### 2.4. Clinical Covariates

At the baseline exam, men self-reported their age, race, marital status, education as well as alcohol and smoking status. Self-report of a physician diagnosis of stroke, diabetes, Parkinson’s disease, chronic obstructive pulmonary disease, myocardial infarction, angina, congestive heart failure and cancer was collected. Participants also self-reported their activity level (Physical Activity Scale for the Elderly, PASE) [31]. The walk speed at the usual pace was measured over a 6-m course using the average of two trials (m/s) [32]. 

### 2.5. Statistical Analysis

We compared characteristics of participants by quintiles of weight using ANOVA and Kruskal–Wallis tests as appropriate. Repeated measures generalized estimating equations (GEE) were used to estimate the odds ratio for mobility limitation across the study contacts (Year 0/baseline, Year 5, Year 7, Year 9, Year 14 and Year 17) for weight, BMI and percent fat (by quintile). Proportional hazards models were used to estimate the hazard ratio and 95% confidence interval for mortality for the weight (by quintile) with follow-up time truncated at 10 years. The proportionality assumption was checked for all models. Splines were used to evaluate the presence of a non-log linear association between weight, BMI or percent fat with mortality or mobility limitation. There appeared to be a non-linear relationship between weight and BMI with mortality; therefore, for mortality models, the referent quintile is the middle (Q3). The association between weight, BMI or percent fat and mobility limitation appeared linear; therefore, the referent quintile is the lowest (Q1). Models were restricted to participants with non-missing data for height, weight, body composition, covariates and mobility limitation at baseline (*n* = 5841, Figure 1). 

We adjusted models for covariates that we a priori considered important potential confounders (but not mediators) of the body size or composition relation with mortality or mobility limitation. These included age, clinical center, race, height (for percent fat and weight models), total energy intake, smoking status and the number of comorbidities. All significance levels were 2-sided. Analyses were conducted using SAS version 9.4 (SAS Institute, Inc., Cary, NC, USA). MrOS data from February 2019 were used (Data can be accessed here: https://mrosonline.ucsf.edu (accessed on 15 January 2020)).

## 3. Results

### 3.1. Characteristics of Participants

Age, race, education, smoking status, walking speed, percent body fat, comorbidity and total energy intake varied across quintiles of weight (Table 1, *p* < 0.05 for all). There was no difference in alcohol use across quintiles of weight. Any difference in marital status across quartiles of weight was of borderline statistical significance. Weight, BMI, percent fat and ALM/ht2 were each highly correlated with one another (r = 0.60–0.87, *p* < 0.001) with the exception of percent fat and ALM/ht2 which were much less strongly interrelated (r = 0.11, *p* < 0.001, Appendix A).

### 3.2. Weight, BMI, Percent Body Fat, ALM/ht2 and Likelihood of Mobility Limitation 

Over an average of 14.4 years, 2802 (48.0%) men reported a mobility limitation for at least one contact/visit, including 800 (13.7%) at Year 1, 778 (17.6%) at Year 5, 735 (19.3%) at Year 7, 1152 (28.8%) at Year 9 and 805 (34.5%) at Year 14, and 867 (42.5%) at Year 17. After accounting for potential confounders, men within the highest quintile of weight were more than three times more likely to develop a mobility limitation than those in the lowest (referent) quartile (Table 2). 

Spline plots (Figure 2A) showed essentially log-linear associations between weight and mobility limitation. Similar, although somewhat stronger, associations were seen for BMI and mobility limitation and for percent fat and mobility limitation (Table 2 and Figure 2B,C). In contrast, there appeared to be a U-shaped relationship between ALM/ht^2^ and mobility limitation, with those with either the lowest or highest ALM/ht^2^ most likely to develop mobility limitation over time (Table 2 and Figure 2D).

### 3.3. Weight, BMI, Percent Body Fat, ALM/ht^2^ and Risk of Mortality

Over 10 years, 1630 (27.9%) of men died. In multivariate models, men with the lowest (Q1) and highest (Q5) weight were somewhat more likely to die during follow-up than men in the middle quintile of weight (Q3). Spline plots confirmed this U-shaped relationship (Figure 3A). Similar associations were seen for percent fat and mortality (Table 2 and Figure 3C), and the U-shaped relation with mortality was even more pronounced for BMI-mortality and ALM-mortality associations (Table 2 and Figure 3B,D).

## 4. Discussion

Overall, we found different patterns of association between measures of body size and composition with mortality versus the likelihood of developing mobility limitations. We found that men with greater weight, higher BMI or greater percent body fat were more likely to develop mobility limitations during follow-up than men with lower values for these measures. In contrast, we observed a U-shaped relationship between ALM/ht^2^ and development of mobility limitation, such that men with both higher and lower values of ALM/ht^2^ were more likely to develop mobility limitations during follow-up than those with intermediate values. For mortality, we found a U-shaped relationship between all measures of adiposity (BMI, percent fat and weight) and mortality, and between ALM/ht^2^ and mortality, such that men in the highest and lowest values of these measures had the highest risk of death compared to those with intermediate values. While previous studies have reported associations between body size or composition with mortality or mobility limitations [3,4,5,6,7,8,9,18,19,20], few have simultaneously reported associations with both outcomes over similar follow-up periods. Our results suggest that there is not a single weight, BMI or percent fat value that represents both the lowest risk of mortality and also the lowest likelihood for developing mobility limitation in older men. 

Given previous studies demonstrating a U-shaped relationship between weight or BMI with mortality, we expected to observe a U-shaped relationship between BMI (or weight or percent fat) with mortality, especially since all the measures are often used interchangeably as proxy assessments of adiposity. We found similar but not completely overlapping associations between these measures and mortality. In fact, when analyzed as quintiles, there was no significant association of percent fat with mortality, although spline analyses support a modest U-shaped association for this relation. Surprisingly few studies have investigated the relationship between percent body fat and mortality [21,22,23,24]—some have shown non-significant associations, while others have reported a higher risk of mortality for those with higher body fat [24]. Others have categorized high versus low percent fat in combination with high versus low lean (i.e., “sarcopenic obesity”), making the referent group different in those studies than the present paper, making it difficult to compare results [21,22]. 

Most previous studies of the relation between ALM/ht^2^ and mortality or mobility limitation have only considered only log–linear (rather than a U-shaped association) with these outcomes [25]. Our data suggest that both low and high values of ALM/ht^2^ may be detrimental. Further, some have suggested that lean mass (as measured by DXA) may explain the U-shaped relation between BMI and mortality. This assertion is supported by analyses that demonstrate that adjustment of the BMI-mortality association for ALM/ht2 attenuates the association between high BMI and mortality [33]. However, given the high correlation between ALM/ht^2^ and BMI (r = 0.71 in our study), these previously reported models are likely to have been unstable due to variance inflation from the very high collinearity of ALM/ht^2^ and BMI. In the present study, when analyzed individually, the relation of BMI with mortality and ALM/ht^2^ with mortality was very strikingly similar. This may lead some to conclude that BMI is a surrogate measure of muscle mass, as DXA lean mass is often used as an approximation of muscle mass [34]. However, lean mass by DXA is not a direct measure of muscle mass [35] and is only modestly related to mortality in other projects [22]. In MrOS, a direct measure of muscle mass by d3-creatine dilution is available [36], but only at the Year 14 clinic visit, precluding its inclusion in the present analyses for its associations with 10-year mortality risk. Therefore, the combined and independent associations of muscle and fat with mortality remain topics for ongoing research, which includes direct measures of each body component. 

There was a clear log–linear association between greater weight, BMI and percent fat and the likelihood of developing mobility limitation, with perhaps a stronger association of BMI and percent fat with mobility limitation than between weight and mobility limitation. These results are consistent with previous studies [18,19,20]. If we accept that those with relatively lower weight or BMI (that is within the “normal” range by most clinical guidelines) truly have an increased risk of mortality but no increased risk of mobility limitation, our results suggest a complicated paradigm. The clinical message would be complicated, as there would exist no single “ideal” weight, percent fat or BMI that confers both the lowest risk of mobility limitation and mortality. Further complicating the interpretation of results, we found that those with the highest and lowest levels of ALM/ht^2^ have a similarly increased risk of mobility limitation and mortality. It is uncertain if the so-called “obesity paradox” may explain these somewhat discrepant findings with mortality versus mobility limitation. For example, if those with lower BMI (or percent fat or weight) are in fact sicker due to comorbid illness or another uncontrolled confounding, and therefore have an apparently higher risk of mortality than those with intermediate values of these measures, then it is unclear why this same unresolved confounding processes do not influence likelihood of mobility limitation (and reflect a higher the likelihood of mobility limitation in those with lower values of these adiposity measures). Much more complex analyses examining contemporaneous change in these measures and the risk of subsequent outcomes may be able to disentangle these complicated relationships. These results suggest that there are different health trajectories that vary according to body weight and composition. 

There are many strengths to this study. MrOS is a large, prospective cohort study with detailed phenotyping and long-term follow-up. We used optimal statistical approaches, including repeated measures GEE and spline analyses, to complete our analyses. However, a few limitations must be noted. MrOS is a study of only men, and the cohort is predominantly white. It is unknown whether these findings would generalize to other populations such as women or minority groups. Further, despite the excellent characterization of participants, there may be unresolved confounding that we could not account for in our models. In addition, the MrOS population was mostly overweight and obese; the cut-point for the lowest BMI quintile was 24.2 kg/m^2^. There were very few underweight men (only 5% had BMI ≤22 kg/m^2^). It is unclear whether similar results would be seen in a population with a more balanced distribution, including a higher representation of normal and additional underweight men.

In conclusion, we observed U-shaped relationships between weight, BMI, percent fat and ALM/ht^2^ with mortality. For mobility limitation, there was a clear log–linear relationship between weight, BMI and percent fat with incident mobility limitation, with higher values associated with a greater likelihood of mobility limitation. There was a U-shaped relationship between ALM/ht^2^ and incident mobility limitation. These observational data suggest that there does not exist a single weight, BMI or percent fat value that can represent both the lowest risk of mortality and also the lowest likelihood for developing mobility limitation over time. 

## Figures and Tables

**Figure 1 geriatrics-06-00053-f001:**
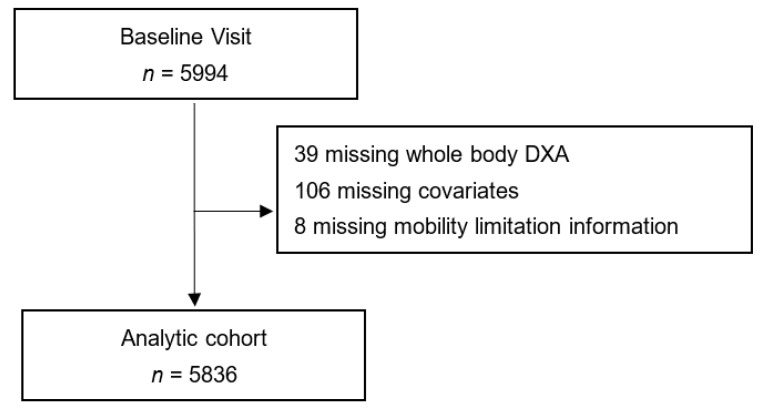
Participant flow chart.

**Figure 2 geriatrics-06-00053-f002:**
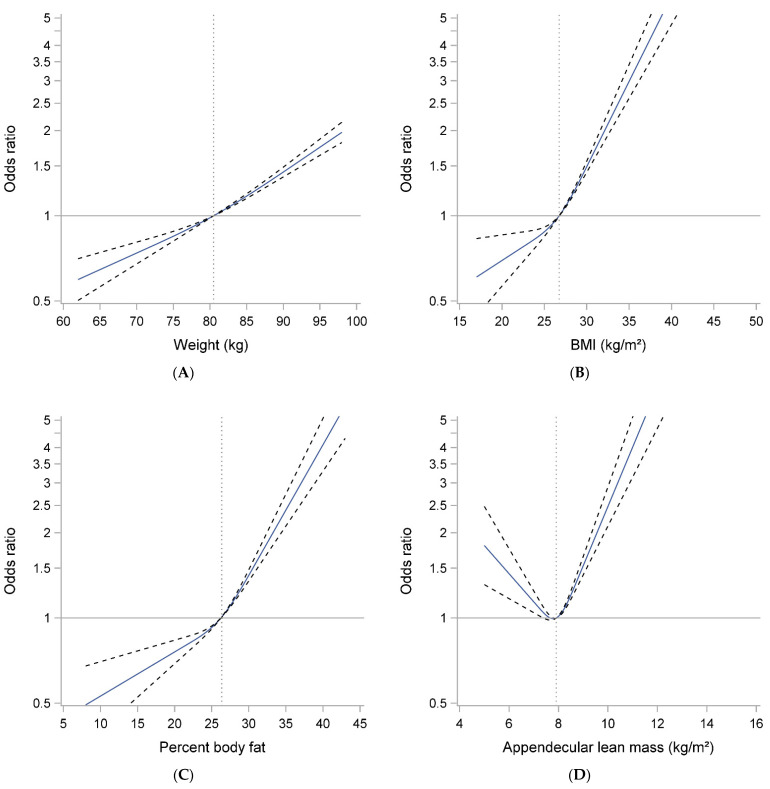
Spline plots of the association between body size and composition ((**A**) Weight, (**B**) BMI, (**C**) Percent Body Fat and (**D**) Appendicular lean mass) with mobility limitation in older men. Models are adjusted for age, clinical site, race, total energy intake, smoking and comorbidity. Weight and percent fat models adjusted additionally for height.

**Figure 3 geriatrics-06-00053-f003:**
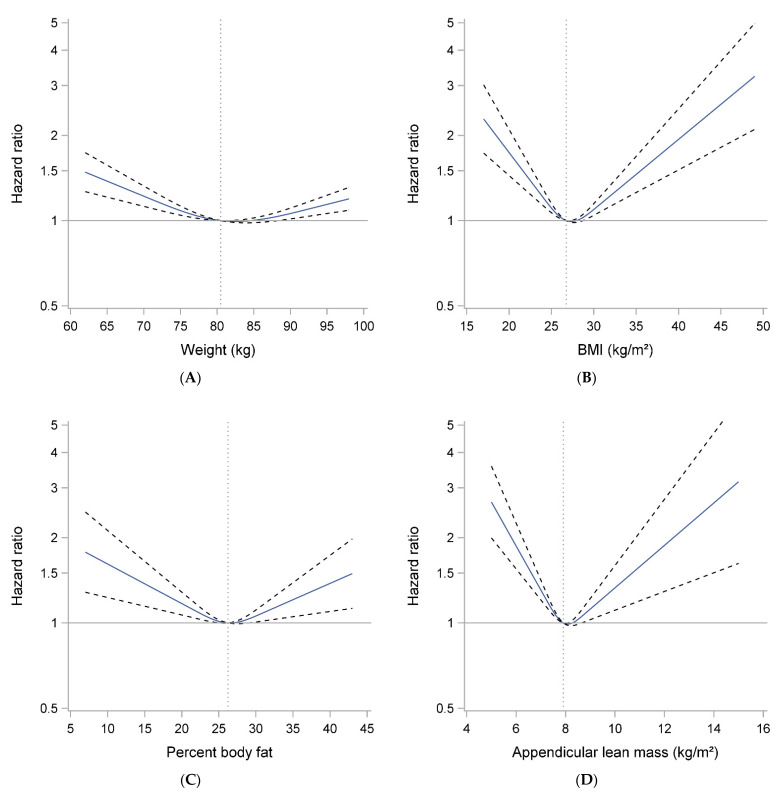
Spline plots of the association between body size and composition ((**A**) Weight, (**B**) BMI, (**C**) Percent Body Fat and (**D**) Appendicular lean mass) and mortality in older men. Models are adjusted for age, clinical site, race, total energy intake, smoking and comorbidity. Weight and percent fat models adjusted additionally for height.

**Table 1 geriatrics-06-00053-t001:** Characteristics of MrOS men free from mobility limitation at the baseline by quintile of baseline weight (kg).

N (%), or Mean ± SD	Baseline Weight (kg) Quintiles					*p*-Value
	Q1:	Q2:	Q3:	Q4:	Q5:	
	N = 1150	N = 1156	N = 1192	N = 1156	N = 1187	
Age, years	76.0 ± 6.5	74.5 ± 5.8	73.3 ± 5.6	72.7 ± 5.4	71.6 ± 4.9	<0.001
Non-Hispanic White	956 (83.1)	1054 (91.2)	1089 (91.4)	1060 (91.7)	1083 (91.2)	<0.001
Married (vs. not)	920 (80.0)	961 (83.1)	1003 (84.1)	951 (82.3)	972 (81.9)	0.105
College educated (vs. < college)	905 (78.7)	908 (78.6)	910 (76.3)	864 (74.7)	871 (73.4)	0.007
Ever smoker (vs. never)	636 (55.3)	684 (59.2)	765 (64.2)	765 (66.2)	800 (67.4)	<0.001
Drinks per day, >2 (vs ≤2)	127 (11.1)	126 (10.9)	151 (12.7)	135 (11.7)	138 (11.7)	0.685
Height, m	1.69 ± 0.1	1.72 ± 0.1	1.74 ± 0.1	1.77 ± 0.1	1.79 ± 0.1	<0.001
Weight, kg	66.5 ± 4.4	75.3 ± 1.9	81.7 ± 1.9	88.9 ± 2.4	102.9 ± 8.5	<0.001
Body mass index, kg/m^2^	23.3 ± 1.9	25.4 ± 1.7	27.0 ± 1.8	28.6 ± 2.0	32.4 ± 3.4	<0.001
Percent body fat	21.9 ± 4.8	24.5 ± 4.3	25.9 ± 4.3	27.7 ± 4.3	31.0 ± 4.5	<0.001
ALM/height^2^ (kg/m^2^)	7.2 ± 0.7	7.6 ± 0.7	8.0 ± 0.7	8.2 ± 0.7	8.9 ± 0.9	<0.001
Comorbidities						0.003
0	340 (29.6)	343 (30.0)	336 (28.2)	308 (26.6)	293 (24.7)	
1	396 (34.3)	405 (35.0)	386 (32.4)	432 (37.4)	394 (33.2)	
2+	414 (36.0)	408 (35.3)	470 (39.4)	416 (35.9)	500 (42.1)	
PASE score	144.6 ± 70.4	145.7 ± 65.1	152.3 ± 68.9	148.0 ± 66.4	143.8 ± 69.1	0.019
Walk speed (m/s)	1.18 ± 0.2	1.22 ± 0.2	1.22 ± 0.2	1.21 ± 0.2	1.18 ± 0.2	<0.001
Total energy intake (kcal)	1545 ± 574	1553 ± 546	1611 ± 611	1685 ± 674	1746 ± 723	<0.001
Incident obility limitation	435 (37.8)	511 (44.2)	547 (47.3)	591 (51.1)	718 (60.5)	0.003
Died during follow-up	427 (37.1)	321 (27.8)	280 (23.5)	293 (25.4)	303 (25.5)	<0.001

**Table 2 geriatrics-06-00053-t002:** The likelihood of self-reported mobility limitation over 14.4 years or the risk of mortality over 10 years by quintile of weight, BMI or percent fat at the baseline exam for MrOS men.

	Mobility Limitation (N = 2802 of 5841)	Mortality (N = 1630 of 5849)
Weight (kg)		
Quintile 1 (<70.7 kg)	1.00 (referent)	1.45 (1.23, 1.70)
Quintile 2 (70.7–77.3 kg)	1.48 (1.27, 1.72)	1.11 (0.94, 1.70)
Quintile 3 (77.3–84.0 kg)	1.61 (1.38, 1.87)	1.00 (referent)
Quintile 4 (84.0–92.4 kg)	2.15 (1.83, 2.51)	1.15 (0.98, 1.35)
Quintile 5 (>92.4 kg)	3.60 (3.04, 4.25)	1.29 (1.09, 1.53)
BMI (kg/m^2^)		
Quintile 1 (<24.0 kg/m^2^)	1.00 (referent)	1.32 (1.13, 1.540)
Quintile 2 (24.0–<25.9 kg/m^2^)	1.23 (1.06, 1.42)	1.03 (0.88, 1.20)
Quintile 3 (25.9–<27.8 kg/m^2^)	1.31 (1.13, 1.52)	1.00 (referent)
Quintile 4 (27.8–<30.2 kg/m^2^)	1.68 (1.46, 1.94)	0.95 (0.81, 1.13)
Quintile 5 (≥30.2 kg/m^2^)	2.83 (2.45, 3.27)	1.30 (1.12, 1.53)
% body fat		
Quintile 1 (<21.8)	1.00 (referent)	1.07 (0.92, 1.25)
Quintile 2 (21.8–<24.9)	1.07 (0.93, 1.25)	0.94 (0.80, 1.10)
Quintile 3 (24.9–<27.6)	1.19 (1.03, 1.38)	1.00 (referent)
Quintile 4 (27.6–<30.7)	1.34 (1.15, 1.56)	0.92 (0.79, 1.07)
Quintile 5 (≥30.7)	1.86 (1.57, 2.20)	1.03 (0.88, 1.19)
ALM/ht^2^		
Quintile 1 (<7.21 kg/ht^2^)	1.00 (referent)	1.38 (1.18, 1.60)
Quintile 2 (7.21–<7.70 kg/ht^2^)	0.95 (0.82, 1.10)	1.04 (0.88, 1.22)
Quintile 3 (7.70–<8.13 kg/ht^2^)	0.93 (0.81, 1.08)	1.00 (referent)
Quintile 4 (8.13–<8.71 kg/ht^2^)	1.05 (0.90, 1.21)	1.01 (0.85, 1.19)
Quintile 5 (≥8.71 kg/ht^2^)	1.56 (1.35, 1.81)	1.15 (0.97, 1.36)

Model adjusted for age, clinical site, race, total energy intake, smoking and comorbidity. Weight and percent fat models adjusted additionally for height.

## Data Availability

Data for MrOS is available at https://mrosonline.ucsf.edu/ (accessed on 15 January 2020).

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
