# Peer review of "Body Weight, BMI, Percent Fat and Associations with Mortality and Incident Mobility Limitation in Older Men"

_geriatrics, 2021, doi:10.3390/geriatrics6020053_

Round 1
Reviewer 1 Report
This is an interesting and well written paper on the important issue of the effect of body mass and adiposity on mortality and mobility limitation in older men. The aim of this article is significant, especially in the context of the increasing prevalence of obesity. Moreover, it is well known that obesity is a significant factor contributing to the increased risk of many serious diseases and mortality, although results on elderly people are not entirely conclusive. Therefore, research on this issue are still needed. Authors revealed that there is not a single weight, BMI or percent fat value that represents both the lowest risk of mortality and also the lowest likelihood for developing mobility limitation in older men. The great advantage of this study is that it is based on the large prospective cohort study with detailed phenotyping and long term follow-up.
Introduction is very well written, concise and comprehensive. The Authors conducted a broad literature review. It is clear what is already known and what needs to be investigated. The rationale and purpose of the research are also clearly formulated.
Some minor comments to the introduction section:
Authors write: “when these factors are taken into account, obesity or overweight have no longer been associated with higher rates of mortality” (Bowman et al. 2017) – I think some error has crept in here because Bowman et al. 2017 wrote: “Obesity is associated with shorter survival plus higher incidence of coronary heart disease and type 2 diabetes in older populations after accounting for the studied confounders”
The materials and methods section precisely describes sample included into the study, measurements taken, and methods applied. The research was carried out using appropriate tools and methods, including statistics.
Minor comment:
In my opinion, Figure 1 seems redundant, this information seems not essential enough to be included in a separate figure, rather it would be sufficient if this information on the number of participants with missing data appears in the text.
The results section clearly shows the obtained outcomes. Appropriate statistical methods were applied and important potential confounders were controlled.
In the discussion section, the Authors clearly present the obtained results and compare them with others previously carried out on this topic. The authors also indicate the advantage of their results over previously published works, but also honestly point to the limitations, which, however, do not seem to bias the obtained results.
Minor comments: 4th paragraph (in parentheses): “and reflect a higher likelihood of mobility limitation in those with lower values of these adiposity measures” – I get what authors had in their minds, but it is confusing, as it, actually, reflects a lower likelihood of mobility limitation, therefore, maybe it would be more clear, if this this part of the sentence would be rewritten somehow, e.g. by adding “do not” before “reflect”?
The conclusions section comprehensively summarizes the results obtained by the authors.
Based on the above, and considering importance of the presented results, I suggest to accept the paper to be published in the Geriatrics.
Author Response
Reviewer #1
Comment #1:
This is an interesting and well written paper on the important issue of the effect of body mass and adiposity on mortality and mobility limitation in older men. The aim of this article is significant, especially in the context of the increasing prevalence of obesity. Moreover, it is well known that obesity is a significant factor contributing to the increased risk of many serious diseases and mortality, although results on elderly people are not entirely conclusive. Therefore, research on this issue are still needed. Authors revealed that there is not a single weight, BMI or percent fat value that represents both the lowest risk of mortality and also the lowest likelihood for developing mobility limitation in older men. The great advantage of this study is that it is based on the large prospective cohort study with detailed phenotyping and long term follow-up.
RESPONSE: Thank you for your careful review of this paper. For brevity, we have only included responses to criticisms raised by the reviewer (and we thank the reviewer for the compliments about our paper).
Authors write: “when these factors are taken into account, obesity or overweight have no longer been associated with higher rates of mortality” (Bowman et al. 2017) – I think some error has crept in here because Bowman et al. 2017 wrote: “Obesity is associated with shorter survival plus higher incidence of coronary heart disease and type 2 diabetes in older populations after accounting for the studied confounders”
RESPONSE: We apologize for our clumsy attempt to summarize the complex analyses of Bowman. The text has been correct to read:
“However, this apparently paradoxical association may be explained by the confounding influences of smoking and involuntary disease-associated weight loss. In fact, when these factors are taken into account, obesity or overweight no longer have the seemingly paradoxical association where obesity is a risk factor for mortality; instead, after accounting for confounders, obesity is associated with higher rates of mortality.9”
In my opinion, Figure 1 seems redundant, this information seems not essential enough to be included in a separate figure, rather it would be sufficient if this information on the number of participants with missing data appears in the text.
RESPONSE: We have omitted Figure 1 from the report.
Minor comments: 4th paragraph (in parentheses): “and reflect a higher likelihood of mobility limitation in those with lower values of these adiposity measures” – I get what authors had in their minds, but it is confusing, as it, actually, reflects a lower likelihood of mobility limitation, therefore, maybe it would be more clear, if this this part of the sentence would be rewritten somehow, e.g. by adding “do not” before “reflect”?
RESPONSE: Thank you for the careful reading of this section. We have updated the text as recommended.
Reviewer 2 Report
- The term ‘percent fat’ is confusing and is best replaced by ‘percentage of body fat’. Percent fat can be places in between brackets, especially in the title.
- The abstract gives a poor reflection of the study. The second sentence is rather confusing.
- The term mobility limitation is best replaced by limitation of mobility.
- what does the term ‘community dwelling’ mean? Please define.
- How long is a ‘block’ roughly in metres.
- Page 1, sentence 37; is it appropriate to use the verb operationalise?
- Pages 1-2, lines 44-48. The long sentence is not very clear.
- The manuscript presents important and interesting findings but reading it several times, I found rather heavy reading and not flowing smoothly.
Author Response
Response to Reviewer #2
The term ‘percent fat’ is confusing and is best replaced by ‘percentage of body fat’. Percent fat can be places in between brackets, especially in the title.
RESPONSE: We struggled with finding the right terminology for this metric. We have changed the language from “percent fat” to “body fat percentage” as this wording was clear in all uses in the paper.
The abstract gives a poor reflection of the study. The second sentence is rather confusing.
RESPONSE: We have revised the abstract to improve clarity.
The term mobility limitation is best replaced by limitation of mobility.
RESPONSE: We respectfully disagree. The term “mobility limitation” is used consistently throughout the literature. Thus, to maintain consistency with previously published papers by us and others, we will continue to use “mobility limitation” throughout.
what does the term ‘community dwelling’ mean? Please define.
RESPONSE: This means that participants were not institutionalized. That is, participants were not residing a nursing home, hospital, incarcerated or other institutionalized facility at the time of recruitment. We have clarified this in the text:
“In 2000-2002, MrOS initially enrolled 5,994 community-dwelling men (i.e., men not residing a nursing home or hospital; not incarcerated; or residing in another institution). Men who were aged 65 years and older, able to walk without the assistance of another person, and free from bilateral hip replacements were recruited at 6 academic medical centers in the United States as described previously..”
How long is a ‘block’ roughly in metres.
RESPONSE: We have clarified this in the text:
“The distance of 2-3 blocks is defined as roughly one-quarter mile or 400 meters.”
Page 1, sentence 37; is it appropriate to use the verb operationalise?
RESPONSE: We agree that our word choice is awkward. We have updated the sentence to read:
“Weight, body mass index (BMI) and body fat percentage are inter-related measures that
reflect overall body size and adiposity and are used to analyze obesity.”
Pages 1-2, lines 44-48. The long sentence is not very clear.
The manuscript presents important and interesting findings but reading it several times, I found rather heavy reading and not flowing smoothly.
RESPONSE: We agree that the language was confusing. The text has been updated to read:
“Some studies,[3-7] but not all,[8,9] have demonstrated a protective effect of overweight and obesity on mortality and other health outcomes in older adults. Other studies have demonstrated a U-shaped relationship with mortality, such that the highest risks in are found for those with either the lowest or the greatest BMI.”